# Nanostructured Non-Ionic Surfactant Carrier-Based Gel for Topical Delivery of Desoximetasone

**DOI:** 10.3390/ijms22041535

**Published:** 2021-02-03

**Authors:** Parinbhai Shah, Benjamin Goodyear, Nirali Dholaria, Vinam Puri, Bozena Michniak-Kohn

**Affiliations:** 1Department of Pharmaceutics, Ernest Mario School of Pharmacy, Rutgers, The State University of New Jersey, Piscataway, NJ 08855, USA; shah.parin@rutgers.edu (P.S.); benjamin.goodyear@rutgers.edu (B.G.); nirali.dholaria@rutgers.edu (N.D.); vinam.puri@rutgers.edu (V.P.); 2Center for Dermal Research, Life Science Building, Rutgers, The State University of New Jersey, Piscataway, NJ 08854, USA

**Keywords:** desoximetasone, corticosteroid drug delivery, controlled drug delivery, niosomal gel formulation, topical drug delivery, skin permeation, allergic reactions, eczema, psoriasis

## Abstract

Psoriasis is a chronic autoimmune skin disease impacting the population globally. Pharmaceutical products developed to combat this condition commonly used in clinical settings are IV bolus or oral drug delivery routes. There are some major challenges for effectively developing new dosage forms for topical use: API physicochemical nature, the severity of the disease state, and low bioavailability present challenges for pharmaceutical product developers. For non-severe cases of psoriasis, topical drug delivery systems may be preferred or used in conjunction with oral or parenteral therapy to address local symptoms. Elastic vesicular systems, termed “niosomes”, are promising drug delivery vehicles developed to achieve improved drug delivery into biological membranes. This study aimed to effectively incorporate a corticosteroid into the niosomes for improving the drug bioavailability of desoximetasone, used to treat skin conditions via topical delivery. Niosomes characterization measurements were drug content, pH, spreadability, specific gravity, content uniformity, rheology, and physicochemical properties. Formulations used a topical gelling agent, Carbomer 980 to test for in vitro skin permeation testing (IVPT) and accelerated stability studies. The developed niosomal test gel provided approximately 93.03 ± 0.23% to 101.84 ± 0.11% drug content with yield stresses ranging from 16.12 to 225.54 Pa. The permeated amount of desoximetasone from the niosomal gel after 24 h was 9.75 ± 0.44 µg/cm^2^ compared to 24.22 ± 4.29 µg/cm^2^ released from the reference gel tested. Furthermore, a drug retention study compared the test gel to a reference gel, demonstrating that the skin retained 30.88 ng/mg of desoximetasone while the reference product retained 26.01 ng/mg. A controlled drug release profile was obtained with a niosomal formulation containing desoximetasone for use in a topical gel formulation showing promise for potential use to treat skin diseases like psoriasis.

## 1. Introduction

The human integumentary system is the most readily accessible organ system and contains the skin, which is the largest human organ serving the purpose of protecting and isolating the internal organs while also being the focus of some novel pharmaceutical dosage forms for drug delivery [1,2,3,4]. Most people will require pharmaceutical intervention for skin diseases and infections at least once in their expected lifetime [5].

Psoriasis and eczema are some of the most common dermatological conditions requiring treatment in patients [6]. The former is a distressing, immune-mediated, chronic inflammatory disorder. The latter is a chronic inflammatory disease of the pilosebaceous unit, having a significant negative impact on patients’ quality of life [7,8,9,10,11]. Psoriasis may appear on the skin in individuals of any age and data suggests that 2–3% of the global population is affected by psoriasis, requiring patients to have effective treatments requiring long-term dosing regimens [12,13]. Psoriasis manifests clinically as excessive growth of the stratum corneum (SC) cells with epidermal thickening, which often appears scaly on the surface with hyperkeratosis, itching, and painful, inflamed skin lesions [9,14,15]. Psoriasis may be present in several forms, including plaque psoriasis, guttate psoriasis, inverse psoriasis, pustular psoriasis, and erythrodermic psoriasis [15,16]. In general, the disease progression and stages are mild, moderate, and severe. While developing effective treatments based on the severity of this disease, several factors need to be considered and include the following: patient age, patient sensorial preferences, adverse side effects, and patient access [17]. In 2018, the estimated annual cost for psoriasis therapies in the United States was 32.5 billion USD [18]. Over 70% of patients with psoriasis prefer a topical product for treating mild to moderate psoriasis due to localization of therapeutic effects, and these formulations are considered to be more effective than those administered by other routes. Topical corticosteroids (CS) are the mainstay of psoriasis treatment [10,13,19] and are available to treat the skin and provide anti-inflammatory, antiproliferative, and immunosuppressive outcomes [17,20].

The topical route for pharmaceutical drug delivery is more acceptable and accessible and now improved due to advancements in technologies that optimize effective delivery. Administration of drugs onto the skin can be divided into two main classes: topical and transdermal. Topical formulations deliver drugs into the layers of the skin. Alternatively, transdermal formulations help to deliver drug molecules much deeper into the dermis for uptake into systemic circulation. Topical dosage forms are widely used for localized therapy and accepted for their effects at the site of administration onto the skin [21,22]. If the skin disease also has systemic manifestations that need to be treated, transdermal drug delivery could be considered or another route of administration could be used, for example, oral or intravenous. Many factors such as the site of action, physicochemical properties of the drug, structure of the drug, etc., play an essential role in designing dosage form delivery [2].

The drug can be delivered by various semisolid dosage forms such as creams, ointments, gels, lotions, suspensions, foams, etc. Among the various semisolid dosage forms, gels are a widely accepted class of dosage form and less tacky compared to ointments and creams [23]. A gel consists of suspended particles in a dispersion medium with a gelling agent that mainly provides stiffness/viscosity to a dispersion which undergoes a high degree of cross-linking or association when hydrated, forming an interlaced three-dimensional structure [24]. Gels tend to be shear-thinning, which results in efficient spreading on the skin with applied pressure or friction.

Topical CS offer convenience to relieve inflammatory responses and are available in various types and doses and in many pharmaceutical dosage forms [20]. Per Stoughton–Cornell classification, CS potencies are defined in seven groups (listed in descending order) according to their vasoconstrictive properties on the skin in vivo [13,19]: super potent (USA class I), highly potent (USA class II/III), moderately potent (USA class IV/V), and low potency (USA class VI/VII) [19]. Even though dermatologists typically prescribe topical CS as a first-line treatment for mild to moderate psoriasis, the side-effects associated with these treatments are cutaneous atrophy, development of striae, rebound of psoriasis, and formation of telangiectasia [19,20]. The dose and applications are important to minimize side effects of CS. For example, the application of CS formulations containing mometasone furoate and fluticasone propionate taken once daily produced less systemic absorption with identical efficacy if compared to twice daily dosing regimens [25,26].

Desoximetasone is a synthetic topical corticosteroid used to treat mild to moderate psoriasis effectively. Desoximetasone has been used in various semisolid dosage forms such as creams, ointments, and gels; however, all of these formulations are limited in terms of controlled release applications, since no product with sustained release is currently commercially available [27,28,29].

In the past few decades, many novel drug delivery systems (many with drug carriers) have been developed and extensively studied to ensure adequate localization and penetration of the drug into the skin with a prolonged therapeutic effect at the site of action [4,30,31]. These novel drug delivery carriers aim to improve the penetration and dermal availability of drugs and provide better patient acceptability by minimizing systemic side effects of the drug. Thus, the treatment of skin diseases using a novel drug delivery system with existing and FDA (Food and Drug Administration) approved drug molecules is preferred [32]. One such innovative topical drug delivery system is a niosome, which are vesicles created by self-assembly of non-ionic surfactants and cholesterol in aqueous media, and can be defined as highly ordered bilayer assemblies of amphiphilic molecules [5,33]. These carriers may be effective at forming drug reservoirs in the upper layers of the skin [31,32]. Today, more than half of the drug candidates demonstrate low aqueous solubility [34], and niosomes can improve the solubility of the poorly soluble drug. These amphiphilic submicroscopic drug transporters enhance the ability to deliver drugs to a target region in a controlled release fashion, to enhance drug deposition in the skin and allow the drug to release from a carrier slowly throughout treatment. Drugs incorporated in niosome carriers for topical drug delivery may demonstrate cellular interactions with epidermal tissues and increase drug residence time in the stratum corneum by releasing the drug in a controlled manner and also reducing systemic side effects [35,36]. By encapsulating desoximetasone in niosomes, the drug release profile can be modulated, which potentially may prevent undesirable side-effects caused by drug accumulation and minimize the need for frequent application of the medication. To provide sufficient contact time with the skin, desoximetasone-loaded niosomes can be incorporated into a gel formulation [37].

This study aimed to investigate the feasibility of topical delivery of desoximetasone-loaded niosomes and incorporation into a pharmaceutical dosage form. Niosomes were developed using an ether injection method by completely dissolving the drug into the organic phase, which was dropwise added to an aqueous phase using magnetic stirring and finally stored separately in glass storage sample containers. Identified during previous studies, Critical material attributes (CMAs) organic phase, drug concentration, surfactant concentration, cholesterol concentration, and lipid types, and critical processing parameters (CPPs) external phase temperature, external phase volume, internal phase volume, mixing speed, mixing time, and rate of addition utilizing a systematic 2^5^ full factorial design using advanced statistical device JMP^®^-enabled DoE approach [6,38]. Formulation composition: containing drug/surfactant/cholesterol (1:2:1), diethyl ether/methanol (75:25), external phase temperature (65 °C), external phase volume/internal phase volume (2:1), mixing speed (650 rpm), mixing time (50 min), and addition rate (0.5 mL/min) successfully developed a final niosome formulation with 90.12 ± 0.02% entrapment efficiency, 449.40 ± 29.2 nm particle size, 0.272 ± 0.03 PDI and −73.50 ± 0.87 mV zeta potential. The topical gel formulation containing Carbomer 980 was optimized using standard reverse engineering and compared with Reference Listed Drug (RLD), identified using FDA Orange Book for identification of the marketed product Topicort^®^ Gel (Taro Pharma) [39].

The optimized topical gel formulation and RLD were characterized using in vitro permeation testing (IVPT) on human cadaver skin to confirm drug release profiles. Additionally, The gel formulation was evaluated for the following: assay for drug, pH, spreadability, specific gravity, color, phase separation, texture, and homogeneity.

## 2. Materials and Methods

### 2.1. Materials

Desoximetasone was received as a gift from Flavine North America, Inc., Closter, NJ, USA. Diethyl ether, stearyl amine, HPLC water, chloroform, calcium chloride dihydrate, docusate sodium, and hydroxypropyl methylcellulose (HPMC) were purchased from Sigma-Aldrich, Saint Louis, Missouri, USA. Ethanol was procured from Decon Labs, Inc., King of Prussia, PA, USA. Acetone, methanol, and acetonitrile were purchased from BDH VWR Analytical, Radnor, PA, USA. Cholesterol and sorbitan monostearate (Span 60) were courtesy of Croda Inc., Mill Hall, PA, USA. Stearic acid was received from BASF Corporation, Edison, NJ, USA. Glacial acetic acid purchased from Fisher Scientific, Fair Lawn, NJ, USA. Edetate disodium, trolamine (TEA), and xanthan gum provided by Spectrum Chemical, New Brunswick, NJ, USA. Ethylcellulose (EC) and hydroxypropyl cellulose (HPC) were courtesy of Ashland Specialty Ingredients, Parlin, NJ, USA. Transcutol was courtesy of Gattefosse Corporation, Paramus, NJ, USA. Carbomer 940, Carbomer 974P, Carbomer 980, Carbomer 981, and Carbomer 1342 were courtesy of Lubrizol Advanced Materials Inc., Brecksville, OH, USA.

### 2.2. Topical Gel Preparation

Purified water was accurately weighed into a glass beaker. An overhead propeller mixer was used to dissolve edetate disodium, docusate sodium, and diethylene glycol monoethyl ether in distilled water to create a complete mixture. Once dissolved, each thickening agent was carefully added to the mix while overhead stirring. In a separate glass container, a ratio of purified water and trolamine was thoroughly mixed using a spatula. The resulting diluted trolamine mixture was slowly added into the mix with overhead stirring to form a preliminary mixture. In a final step, each desoximetasone niosomal dispersion [6] was accurately weighed and added into the previously prepared viscous mixture and mixed using the propeller mixer. Topical gel formulation details are provided in Table 1.

### 2.3. Topical Gel Chemical Evaluation

#### 2.3.1. HPLC Method

HPLC mobile phase was prepared by mixing 65:35:1 ratio of methanol, HPLC-grade water, and glacial acetic acid [40]. The diluent was prepared in a 100 mL volumetric flask by combining 1.50 g of calcium chloride dihydrate into 5.0 mL of HPLC-grade water. This mixture was agitated until calcium chloride dihydrate was completely dissolved. The resulting mixture was made to the final volume using methanol. The drug concentrations were determined using a Discovery C18 column (Sigma-Aldrich, Saint Louis, MO, USA) with 5 µm particle size, L × ID 150 mm × 4.6 mm [40]. Flow rate was set to isocratic 1.50 mL/min, and sample injection volume selected was 20 µL. Sample run time was set to 10 min with a column temperature of 30 °C. The expected analyte retention time (RT) for the drug peak was at approximately 4.0 min. Desoximetasone drug quantification was performed using a validated HPLC method on an Agilent 1100 series (Agilent Technologies, Santa Clara, CA, USA) coupled with UV detection (DAD) at a wavelength λmax 254 nm and HP ChemStation software v. 32.

#### 2.3.2. Desoximetasone Assay Characterization

Desoximetasone niosomal topical gel sample preparation was performed using a silicone spatula and, once well mixed, was added into mobile phase diluent, followed by gentle vortexing to ensure adequate mixing. The resulting sample was sonicated and exposed to 60 °C for 12 min and was then left for gradual cooling at room temperature. Samples were further diluted, and drug quantification was determined using the HPLC method described in Section 2.3.1.

#### 2.3.3. Content Uniformity Measurement

Desoximetasone niosomal topical gel samples were collected for content uniformity measurements from different locations in the container. Sample aliquots were collected from the top, middle, and bottom regions of the gel container. Samples were dissolved by weighing the sample into sample diluent and were then vortexed. The resulting mixture was sonicated for exactly 12 min at 60 °C followed by a gradual cool-down step to allow each sample to equilibrate back to room temperature conditions. The resulting sample was further diluted to prepare for HPLC sample injections, and the drug was quantified using the validated HPLC method, described in Section 2.3.1.

### 2.4. Topical Gel Physical Characterization

#### 2.4.1. pH Measurement

The pH of various desoximetasone niosomal topical gel formulations was determined by mixing 1 g of niosomal gel in 10 g of DI water, using a calibrated pH meter (VWR pH meter symphony B10P, Radnor, PA, USA) at room temperature.

#### 2.4.2. Spreadability Measurement

A 100 mg sample was carefully placed in the center of a microscopic glass slide and covered with another to promote even sample distribution. An additional calibration weight of 50 g was placed on the microscope slide for 1 min until the sample had evenly spread. Spreadability was measured and resulting diameter was recorded in millimeters (mm).

#### 2.4.3. Specific Gravity Measurement

Specific gravity was measured using a metal pycnometer. The pycnometer was rinsed and filled with purified water, and its weight was measured. Then, the pycnometer was emptied, filled with the product, and weighed. The specific gravity of the product is the quotient obtained by dividing the weight of the product by that of water at room temperature as shown in the Equation (1) [41]:(1)$$Specific\;Gravity=\frac{Weight\;of\;the\;product\;\left(g\right)}{Weight\;of\;water\;\left(g\right)}$$

#### 2.4.4. Rheological Evaluation

Rheology is the study of how materials deform and flow from externally applied forces in a controlled way [42]. Rheological evaluations were conducted using a TA Rheometer (Model: Discovery HR-1, Newark, DE, USA) equipped with a 40 mm parallel plate. Method parameters were defined as 25 °C with a testing gap of 1000 µm, loading gap of 45,000 µm, and trim gap offset of 50 µm.

(1) Yield Stress Measurement

The sample temperature was set to 25 °C with a 60 s soak time subject to an increasing shear rate of 0 to 1000 1/s for 120 s with a sampling interval of 2 s/pt. A resulting graph was selected to plot shear rate ẏ (1/s) against shear stress σ (Pa).

(2) Flow Curve Measurement

During flow experiments, increasing shear stress forces are plotted based on the method parameters, followed by an inversely proportional decrease. A resulting shear rate is measured by producing a flow curve. The testing temperature was 25 °C and the soak time of the sample was 60 s. This test was performed in two steps, the initial step for the upward curve involved subjecting the samples to an increasing shear rate of 0 to 1000 1/s for 120 s, and the second step for the downward curve—decreasing shear rate of 1000 to 0 1/s for 120 s. The sampling interval was selected 2 s/pt., and curves with shear rate ẏ (1/s) vs. stress σ (Pa) were obtained.

(3) Viscosity Measurement

Viscosity was measured at 25 °C with a soak time of 60 s. The samples were subjected to an increasing shear rate of 0 to 1000 1/s for 120 s with a sampling interval of 2 s/pt. The shear rate ẏ (1/s) vs. viscosity η (Pa.s) curves obtained allowed determination of the viscosity at a low, medium, and high shear rate.

### 2.5. Evaluation of Physicochemical Property

#### 2.5.1. Color, Texture, and Phase Separation

Topical gels were visually evaluated for color, phase separation, and gel surface texture.

#### 2.5.2. Homogeneity

Qualitative determination of the niosomal gel was evaluated by placing the gel between the thumb and the index finger, and sample homogeneity or the presence of aggregates were observed.

### 2.6. Selection and Optimization of Thickening Agent

#### 2.6.1. Selection of Thickening Agent

The gelling agent is a crucial excipient needed to form a pharmaceutical gel formulation adequately and must be optimized to achieve targeted viscosity profiles of the final product matching the RLD. Carbomer 940 has been identified as a gelling agent in the Topicort^®^ gel, 0.05% formulation manufactured and distributed by Taro Pharmaceuticals, Hawthorne, NY, USA [43]. Carbomer 940 is not recommended for use in topical formulations due to drawbacks, including high benzene content recognized by the FDA as a carcinogen. The manufacturer suggests Carbomer 980 as a non-benzene substitute [44], since they both contain very similar physical and chemical properties. Thus, Carbomer 980 was selected as the excipient of choice for gel formation due to its relatively low systemic side-effect risk.

#### 2.6.2. Optimization of Thickening Agent Concentration

Thickening agents require sufficient optimization to achieve the target viscosity and other physicochemical parameters relevant to formulation performance criteria. Samples are typically compared to associated RLD to confirm that both treatments are bioequivalent, in cases would warrant sufficient evidence to apply for a biowaiver. Formulations manufactured contained 0.62%, 0.70%, 1.00%, 1.50%, and 2.00% Carbomer 980 gelling agents. Samples prepared tested for rheological performance (yield stress and viscosity) measurements for optimizing the amount of gelling agent required in the sample concentration.

#### 2.6.3. Rheological Test Methods, Parameters and Evaluation

The rheological evaluation was performed on a TA Rheometer (Model: Discovery HR-1, Newark, DE, USA) equipped with a 40 mm parallel plate. All the tests were completed at 25 °C with a testing gap of 1000.0 µm, loading gap 45,000.0 µm, and trim gap offset 50.0 µm. The test temperature was 25 °C and the soak time of the sample was 60 s. The samples were subjected to an increasing shear rate of 0 to 500 1/s for 120 s. The sampling interval selected was 5 s/pt. The same flow-ramp run was used to measure yield stress and viscosity (at the low, medium, and high shear rate) of the sample. For yield stress and viscosity evaluation, the graph was plotted for shear rate ẏ (1/s) vs. stress σ (Pa) and shear rate ẏ (1/s) vs. viscosity η (Pa.s), respectively.

### 2.7. Formulation Selection

Niosomal topical gel formulation is guided by selecting comparable to reference product composition [43], as depicted in Table 1.

### 2.8. Selection of Optimized Desoximetasone Niosomal Gel Formulation

All desoximetasone niosomal topical gel and reference marketed gel formulations were evaluated for their physicochemical properties, which formed the basis of selection of the optimized formulation.

### 2.9. In Vitro Permeation Study Using Human Cadaver Skin

#### 2.9.1. Study Design

In vitro skin permeation studies were executed using Franz Diffusion Cells (FDCs) with an exchange area of 0.64 cm^2^ and receptor volume of 5.0 mL (Permegear Inc., Hellertown, PA, USA). Dermatomed human cadaver skin from the posterior torso of a 47-year-old white male (The New York Firefighters Skin Bank, NY, USA) was slowly thawed at room temperature, cut into appropriate pieces, and soaked in filtered phosphate buffer saline pH 7.4 for 15 min. Then, the skin samples were mounted on FDCs with the epidermal side in contact with the formulations in the donor compartment. The receptor compartment was filled with ethanol/water (40:60) as the receptor media in order to achieve sink conditions and was maintained at 37 °C under continuous stirring at 600 rpm using a magnetic stirrer. After equilibration at 37 °C for 15 min, the test and control formulations were applied in a zigzag-like manner. Three hundred microliters of receptor samples were withdrawn from the sampling port at 1, 2, 3, 4, 6, 8, 10, 12, 16, 20, 22, and 24 h, immediately followed by replenishment with fresh media. The collected samples were analyzed using the validated HPLC method described in Section 2.9.2.

#### 2.9.2. Analytical Testing Parameters

The HPLC instrument used was Agilent 1100 series equipment (Agilent Technologies, Santa Clara, CA, USA) coupled with UV detection (DAD) and HP ChemStation software V. 32. For the analysis of desoximetasone, a mobile phase of 60% methanol and 40% water was pumped through a Discovery C18 column (Sigma-Aldrich, Saint Louis, MO, USA) with 5 µm particle size, L × ID 150 mm × 4.6 mm column. Injection volumes of 20 µL with a flow rate of 1.00 mL/min set to 30 °C with UV detection of 254 nm was used with the retention time of 4 min. The receptor media of the permeation study was utilized as diluent.

#### 2.9.3. Data Analysis

Penetration parameters were obtained from the cumulative amount of desoximetasone permeated per unit skin surface area (µg/cm^2^) versus time (hours) plot.

The cumulative amount of desoximetasone permeated per unit area was calculated according to Equation (2):(2)$$Q_{n}=\frac{C_{n}V_{r}+\sum_{i=0}^{n-1}C_{i}V_{s\;}}{A}$$
where *Q_n_* is the cumulative amount of the drug permeated per unit area (µg/cm^2^) at different sampling times, *C_n_* is the drug concentration in the receiving medium at different sampling times (µg/mL), *C_i_* is the drug concentration in the receiving medium at the ith (*n* − 1) sampling time (µg/mL), *Vr* is the volume of the receptor solution (mL), *V_s_* is the volume of the sample withdrawn (mL), and *A* is the effective permeation area of the diffusion cell (cm^2^). The *Q_n_* values were plotted against time, and the steady-state flux (Jss) was calculated from the slope of the linear portion of the plot. Results are reported as mean ± SD (*n* = 6). The statistical analysis of the data was performed using one-way ANOVA, Tukey’s post hoc test and Student’s *t*-test, and *p*-values < 0.05 were considered significant.

### 2.10. Skin Deposition Study

At the end of the permeation study, the skin was removed from the diffusion cells, cut around the diffusional area, air dried, and weighed accurately. The skin pieces were placed into BeadBug™ tubes, which were then cut in tiny pieces using surgical scissors. Each tube was filled with 1 mL ethanol and homogenized for 3 cycles of 180 s using BeadBugTM Microtube homogenizer D1030 (Benchmark Scientific, Sayreville, NJ, USA). The tubes containing the homogenized skin tissue were then loaded into a Julabo SW22 shaking water bath (Julabo USA Inc., Allentown, PA, USA) set to 37 °C for 24 h. They were then centrifuged for 5 min at 1200 rpm and the supernatant was collected and filtered through a 0.45 µm polypropylene filter into HPLC vials. Finally, the samples were analyzed for drug content using the prevalidated HPLC method as described in Section 2.3.1. Desoximetasone concentrations were expressed as nanograms of desoximetasone per unit skin weight in mg (ng/mg) and are discussed in Section 3.11.

### 2.11. Stability of the Niosomal Topical Gel

The stability of the formulation was evaluated, and it was immediately tested for drug content, pH, spreadability, homogeneity, color, texture, and phase separation at time 0 (initial) sample. The stability test samples (4 g) were stored in capped glass vials at room temperature and 40 °C temperature for 15 days, 1 month, 2 months and 3 months, followed by drug content determination using the methods described in previous sections.

## 3. Results and Discussion

### 3.1. Thickening Agent Concentration Determination

The thickening agent is a critical ingredient defining the viscosity of a gel formulation. It is essential to optimize the thickening agent’s concentration to achieve the targeted viscosity, in this case, to match with the reference product. As previously mentioned, Carbomer 980 was selected as the thickening agent for the test gel due to the similarity observed with Topicort^®^ gel 0.05% based on the rheological evaluation. Yield stress (Figure 1a) and viscosity (Figure 1b) values for test gels containing different concentrations of Carbomer 980 along with the reference gel product are provided in Table 2 and Table 3, respectively.

For yield stress evaluation, shear rate ẏ (1/s) vs. stress σ (Pa) was plotted, and the yield stress value was determined by applying the Herschel–Bulkley model. Yield stress values from Table 2 show that reference marketed gel had a yield stress of 78.97 Pa, and the desoximetasone niosomal gels showed an increase in yield stress as the concentration of Carbomer 980 was increased. Yield stress of 93.44 Pa was observed for the 0.7% Carbomer 980 formulation (Lot# DNTG-5), which was the closest to that of the reference gel.

For viscosity evaluation, the graph was plotted for shear rate ẏ (1/s) vs. viscosity ƞ (Pa.s). Table 3 shows that the reference gel had a viscosity 18.61 Pa.s (~10 shear rate), 2.03 Pa.s (~240 shear rate) and 1.43 Pa.s (~490 shear rate). Viscosity data for the test gels followed a similar concentration-dependent trend as yield stress. The formulation containing 0.7 % Carbomer 980 (Lot# DNTG-5) had a viscosity of 19.96 Pa.s (~ 10 shear rate), 2.02 Pa.s (~ 240 shear rate), and 1.24 Pa.s (~ 490 shear rate), which was the closest value to that of the reference gel. Based on this rheological evaluation, 0.7% Carbomer 980 containing gel formulation (Lot# DNTG-5) was the best match to the reference gel product. Therefore, Lot# DNTG-5 was used for further evaluations, including various thickening agents to compare with the final product.

### 3.2. Desoximetasone Drug Content (Assay) Determination

Drug content from the various topical gels was analyzed along with the reference product. This was done to ensure that an adequate amount of active ingredient was added to the formulations. Drug content values of the niosomal formulations were found in the range from 93.03 ± 0.23% to 101.84 ± 0.11%, and that of the reference marketed gel product was 101.80 ± 0.38% (Table 4). Since all the formulations displayed consistent assay values with little deviations, it could be confirmed that the amount of active added to the formulations was adequate.

### 3.3. pH Measurement Study

The pH of a semisolid product potentially affects the stability of the product’s active ingredient and physicochemical properties. This may have an impact on the effectiveness of added preservatives as well as the viscosity of the drug product. The pH of desoximetasone niosomal gel formulations was dependent on the quantity and type of the thickening polymer. The pH values of the test formulations were found in the range 4.39 ± 0.01%–9.28 ± 0.02% and 5.64 ± 0.01% for the reference gel (Table 4). The data shows the pH of the formulations was well controlled and acceptable.

### 3.4. Spreadability Study

The spreadability of the formulation is inversely influenced by the viscosity. Good spreadability is a critical criterion for a gel formulation, as it depicts the product’s behavior when it comes out from the container. It is used to indicate the extent of the area to which gel could readily spread upon application. The spreadability of the niosomal gels was in the range of 17.00 ± 0.00% to 26.33 ± 0.58% mm, and that of the reference gel was observed to be 18.67 ± 0.58% mm, as shown in Table 4. It was found that the spreadability of the niosomal gels decreased by increasing the thickening polymer concentration and that it changed with a change in the type of thickening polymer.

### 3.5. Specific Gravity Study

The specific gravity of semisolid products is used to verify the presence of excessive amounts of entrapped air in them. It is essential to observe specific gravity, as excess air in the product can lead to spoiling and degradation of the ingredients and affect physical properties such as viscosity. Specific gravity was observed to be in the range of 0.952 ± 0.00% to 1.009 ± 0.00% for the niosomal gels and 0.947 ± 0.00% for the reference gel, as shown in Table 4.

### 3.6. Topical Gel Content Uniformity Study

Content uniformity evaluation is essential in the case of semisolid products that contain the drug in the dispersed phase. The test was conducted by collecting aliquots from the top, middle, and bottom positions of the undisturbed finished product. The content uniformity study data from the desoximetasone niosomal topical gel and reference gel products are provided in Table 5.

Formulations DNTG-1 to DNTG-9, DNTG-13, and the reference gel showed identical content uniformity, whereas formulations DNTG-10 to DNTG-12 did not. This can be explained by the fact that the former group of formulations were semisolids and did not allow the particles to settle, while the latter was in liquid form, causing settling down of the niosomes and thus resulting in higher drug content in the bottom samples compared to the top and middle samples.

These results demonstrate that the gel manufacturing process was efficient in uniformly distributing desoximetasone-loaded niosomes throughout the batch and that the gel matrix was able to hold the niosomes well.

### 3.7. Rheological Study

The rheological studies of gels help understand the semisolid microstructure of the formulated gel product. Microstructure helps formulators to better understand the viscoelastic properties that govern the performance of a gel, including product and process performance, as well as stability. This helps reduce the potential risk of failures by providing details that help predict which formulations might have some potential for flocculation, coagulation, or coalescence, resulting in undesired effects, such as settling, creaming, or separation. Various gel formulations were manufactured by either changing Carbomer 980 concentration or by replacing Carbomer 980 with other thickening agents such as Carbomer 940, Carbomer 974P, Carbomer 981, Carbomer 1342, xanthan gum, hydroxypropyl cellulose (HPC), hydroxypropyl methylcellulose (HPMC), and ethyl cellulose (EC). Rheological yield stress and viscosity data of the various niosomal gels and reference marketed gel are provided in Table 6 [45].

#### 3.7.1. Yield Stress Measurement Study

Yield stress is defined as the minimal stress that must be applied to the sample to disrupt the structure in material, causing sample material to flow. A yield stress value is required to determine the product’s flowability point and is considered a critical quality attribute (CQA) for semisolid product stability. Results demonstrate a strong correlation with increasing the gelling agent concentration with increased yield stress of the product for every type of gel. The graph was plotted for shear rate ẏ (1/s) vs. stress σ (Pa) and yield stress value for all the samples was determined by applying the Herschel–Bulkley model. The Herschel–Bulkley equation can be described as in Equation (3):(3)$$\tau =\tau_{0}+k{\left(\gamma \right)}^{n}$$
where *τ* = shear stress, *τ*_0_ = yield stress, *k* = consistency factor, *γ* = shear rate, and *n* = flow index, a power-law exponent [46].

The yield stress behavior comparison for all the niosomal gels and reference gel drug product are provided in Figure 2 and reveal that all the samples exhibited pseudoplastic flow with a presence of yield stress.

#### 3.7.2. Flow (Upward and Downward) Curve Behavior Study

Flow curve measurement predicts how a semisolid drug product might behave in various applications, from dispensing the formulation from a tube to how the material may behave during scale-up machines dispensing from fill lines. The upward and downward curves give formulation scientists information on sample breakdown and recovery, whether the sample structure recovers immediately when the stress is removed, or does the sample recover over a longer period of time. In use, the formulation structure should disrupt easily. At the same time, the tube or packaging component is squeezed, flow when being applied to the skin, and then rebuild structure quickly (negligible hysteresis) to prevent runoff from the skin. From the flow curve profile of all the formulations shown in Figure 3, it may be interpreted that all the formulations showed non-Newtonian flow behavior (pseudoplastic flow) with yield stress. Additionally, all the formulations’ downward curve followed the same path as the upward curve, which suggests that all the formulations were nonthixotropic.

#### 3.7.3. Viscosity (Low, Medium, and High Shear Rate) Study

Viscosity is a measure of the internal friction of a fluid, the resistance to flow. Viscosity testing requires applying shear stress to sample materials and measuring the resulting rate of flow caused by the addition of stress. Table 6 lists the viscosity values of all the samples at low shear rate (~8 1/s), medium shear rate (~493 1/s), and high shear rate (~991 1/s). For viscosity evaluation, the shear rate ẏ (1/s) vs. viscosity ƞ (Pa.s) plots (Figure 4) show the changes in viscosity as the concentration of the gelling agent decreased and as the shear rate increased, which demonstrated pseudoplastic flow behavior in all samples.

### 3.8. Physicochemical Formulation Properties Evaluation

Table 7 shows the physicochemical properties of prepared desoximetasone niosomal gel formulations DNTG-1 to DNTG-13 and reference gel. Prepared niosomal gels’ appearance may be described as opaque to white color and were comparable to the reference listed gel product. Formulation DNTG-1 to DNTG-9, DNTG-13, and reference gel showed good homogeneity with no lumps and smooth homogeneous texture. DNTG-10 to DNTG-12 samples had poor homogeneity and liquid consistency due to the gelling agents’ intrinsic nature for the formulation. Phase separation was not observed in any gel formulations due to the hydrophilic nature of the gelling agents and ability to interact with the microstructure gel network. Furthermore, no API crystallization nor precipitation were observed with the formulations.

### 3.9. Selection of Ideal Desoximetasone Niosomal Topical Gel Formulation

As mentioned since the beginning of the study, the ideal desoximetasone niosomal gel formulation would be the one that shows the most similarity in characteristics with the reference gel. The objective while developing the new formulations was to get a formulation that matches physical characteristics with the reference product. The only difference was the form in which the API is present. The ideal formulation with the drug entrapped in the niosome matrix may be compared against the reference gel, which had a drug in solubilized form.

As shown in Table 1, thirteen different formulations were made by changing either concentration of Carbomer 980 or the type of the thickening agent to find the most suitable thickening agent for the desoximetasone niosomal gel. These gels were compared with the reference gel product by evaluating various physicochemical characteristics such as drug content, content uniformity, pH, spreadability, specific gravity, color, texture, homogeneity, phase separation, and rheological properties as detailed in Table 4, Table 5 and Table 6. The resulting data demonstrates that formulation’s DNTG-5, which contains 0.7% Carbomer 980, data matches with the reference gel. The comparison between DNTG-5 and H882532755 has been summarized in Table 8. Figure 5a–c shows rheological overlay graphs of the formulations.

Figure 5a displays the complete flow (upward and downward) curve of shear stress versus shear rate of DNTG-5 and H882532755 (*n* = 3). From the flow curve profile of all the samples it can be interpreted that both niosomal gel and reference marketed gel drug products showed non-Newtonian flow behavior (pseudoplastic flow) with yield stress. Additionally, the downward curve for all the sample runs followed the same path as the upward curve, which suggest that both niosomal gel and reference gel drug product were nonthixotropic.

Figure 5b depicts the viscosity profile comparison of DNTG-5 and H882532755 across the range of possible shear rates. For viscosity evaluation, the graph was plotted for shear rate ẏ (1/s) vs. viscosity ƞ (Pa.s). The average viscosity values (*n* = 3) of niosomal gel and reference marketed gel drug product at low (~8 1/s), medium (~493 1/s), and high shear rate (~991 1/s). The average viscosities for the DNTG-5 were 20.82 Pa.s, 1.21 Pa.s, and 0.78 Pa.s, respectively, at low, medium, and high shear rates, and for H882532755 they were 20.95 Pa.s, 1.25 Pa.s, and 0.80 Pa.s, respectively. From Table 8 it can be seen that these averages were obtained for the two types of gels. Additional observations demonstrated that the viscosity for all the samples decreased as shear rate increased, showing pseudoplastic flow behavior in all the samples.

Figure 5c displays the yield stress comparison of DNTG-5 and H882532755 (*n* = 3). The graph reveals that all the samples exhibited pseudoplastic flow with the presence of yield stress. The graph was plotted for shear rate ẏ (1/s) vs. stress σ (Pa), and yield stress value for the samples was measured by applying the Herschel–Bulkley model. The average yield stress values (*n* = 3) obtained for DNTG-5 and H882532755 were 106.57 Pa and 118.88 Pa, respectively.

The results obtained from DNTG-5 (0.70% Carbomer 980) samples contained the ideal desoximetasone gel formulation quality attributes and were considered for further testing.

### 3.10. In Vitro Permeation Study Evaluation

The permeation study was performed between the test (DNTG-5) and reference formulations (H882532755) to evaluate the effect of loading the drug into niosomes on drug release into human cadaver skin and the extent of drug deposition in skin layers over time.

#### 3.10.1. Drug Permeation through Human Cadaver Skin

The cumulative amount of desoximetasone penetrated from the test and reference formulation’s parameters are comparatively illustrated in Table 9. Permeation profiles of desoximetasone after application of test and reference gels, obtained from samples collected at regular intervals (1, 2, 3, 4, 6, 8, 10, 12, 16, 20, 22, and 24 h), including the amount of desoximetasone detected in human cadaver skin after 24 h, are shown in Figure 6.

Results demonstrated that the flux from the reference formulation lot# H882532755 was greater in comparison with the test formulation lot# DNTG-5. Niosomes showed biphasic drug release, an initial relatively faster release phase lasting 4 h followed by a slower release phase extending up to 24 h. During the initial step, the free/unbound or loosely bound drug at the niosomal bilayer’s surface permeated to reach the release medium [47]. In contrast, in the slow-release phase, the entrapped drug leaked out gradually from the niosome vesicles into the medium. The addition of stearic acid into niosomes formulation reduced both release rate and extent of drug release from the niosome formulation through the study period, the effect being concentration-dependent. The sustained release of drug from niosomes was observed due to the release-retarding effect of the bilayer stabilized by cholesterol [48]. An additional explanation of the sustained release drug from the negatively charged niosome vesicles could be retaining the positive drug by the negative bilayers. A similar trend has been established by several previously published observations [49,50].

The drug release study indicates that the release pattern between the reference and test product was identical; however, drug release from the desoximetasone niosomal gel was steady and slower compared to the referenced gel product.

#### 3.10.2. Kinetic Analysis of the Drug Release Data

The first order equation correlates with the release from system, where release rate is concentration-dependent, expressed by the equation:Log *C_t_* = Log *C*_0_ − *kt*/2.303(4)
where *C*_0_ is value of initial drug concentration, *C_t_* is value of drug concentration at time *t*. The data obtained were plotted as log cumulative of % drug remaining vs. time which would yield a straight line with a slope of −*K*/2.303 [51].

The mathematical model targeted to describe drug release from the matrix system was initially proposed by Higuchi in 1963. This model relates the release of drug from insoluble matrix as a square root of time dependent process based on Fickian drug diffusion equation.
*Q* = *KH**t*^1/2^(5)
where *Q* is the amount of drug release in time *t*. The plot made as cumulative percentage drug release vs. square root of time. The slope of the plot gives the Higuchi dissolution constant *KH* [52].

Ritger and Peppas [53], and Korsmeyer and Peppas [54] developed an empirical equation to analyze Fickian as well non-Fickian drug release from swelling and nonswelling polymeric delivery systems. The drug release mechanism calculated using the Korsmeyer–Peppas equation.
*M_t_*/*Mα* = *K t^n^*(6)
where *M_t_/Mα* is the fraction of drug released at time *t* and *k* is the rate constant. *n* is the release exponent indicative of the mechanism of drug transport through the polymer [55].

The first order and Higuchi square root law model evaluated the drug release kinetics from the polymer matrix systems. A correlation obtained from release studies was defined by selecting the best of two standard models for the fit analysis. The correlation coefficient *R*^2^ value of the first order was a better fit for the release profile of H882532755, and DNTG-5’s release profile was a better fit for the Higuchi model (Figure 7). Drug molecules released from the niosomal gel matrix were obtained from an initial and later phase. The first phase (a) is an initial phase where free/unentrapped or loosely bound drug and drug adsorbed at the surface of niosomal bilayer diffuses into the release medium, in agreement with the research conducted by Akbari et al. about the release of ciprofloxacin-loaded niosomes [47]. The later phase (b) is a slow-release phase, where the entrapped drug releases slowly from the niosome vesicles. The addition of charge in the niosomes further reduces both release rate and the extent of drug released through the study duration, the effect being concentration-dependent. The sustained drug release from niosomes was also due to the release-retarding effect of the bilayer stabilized by cholesterol. This is in agreement with Gurrapu’s et al. research on the oral delivery of valsartan from maltodextrin-based proniosome powders [48]. The diffusion exponent of the Korsmeyer–Peppas equation was less than 0.5 (DNTG-5 exponent *n* value 0.3). This indicates that the Fickian mechanism was dominant and controlled the drug release from the DNTG-5 gel formulation. Feith et al. observed similar results with fluconazole-loaded niosomal gels for topical ocular drug delivery for corneal fungal infections [56]. The pharmacokinetics release parameters and correlation coefficients (*R*^2^) calculations for the reference gel and test gel formulations are accurately summarized in Table 10.

### 3.11. Drug Deposition in Human Cadaver Skin

Drug deposition was studied on human skin using in vitro permeation testing (IVPT) during and after the permeation study. Skin deposition study results demonstrated a marked increase in drug release improvement over an extended amount of time. The skin deposition of desoximetasone from the test and reference products is summarized in Table 11 and Figure 8.

It was observed that niosomal gel was able to retain more drugs in human cadaver skin compared to the reference gel product, which might be useful in the treatment and management of skin diseases to reduce the frequency of drug product application.

### 3.12. Stability of Desoximetasone Niosomal Gel

It is essential to monitor the gel product’s stability containing drugs incorporated into the niosome vesicles. Semisolid products may show various kinds of physical instability such as creaming, flocculation, Ostwald ripening, change in viscosity, or rheological behavior either by structure build-up or structure breakdown, change in particle size either by increasing or decreasing particle size. Sedimentation, color change, pH change, or conductivity may also be a problem for stability. The typical examples of chemical instability are reduction in drug assay or increase in degradation products due to the degradation of drug substance following a chemical reaction with other ingredients in the formulation, or environmental instability like thermal, oxidative, or light-induced degradation. Formal stability testing is required to support the product’s shelf life and acceptance with a regulatory submission point of view. The ICH stability testing guidelines [57] provide details on the storage temperatures and testing frequency for different product types, other packaging container-closure type, and intended markets. Suggestions from the FDA regarding the stability guidelines cover physical and chemical attributes that apply to drug products.

Similar to other semisolid products such as creams, ointments, and lotions, topical gels can deliver the drug at the target site, and drug release may be modified by changing the viscosity-defining polymer. However, to be a commercially viable treatment option, topical gel and the drug(s) within must be stable for extended periods. Typically, a shelf life of two years is advised for such formulations. Drug content, pH, spreadability at room temperature and 40 °C temperature are presented in Table 12 and depicted in Figure 9a–c, respectively.

Table 13 shows the physicochemical observations of various stability samples of desoximetasone niosomal gel stability batch.

There were no significant observable changes in the stability samples regardless of time and temperature. Results demonstrate that desoximetasone niosomal gel formulations remained stable over a longer amount of time and at various temperature ranges. Further evaluations are needed to understand the nuances of microstructural differences occurring with niosomal vehicles, which can be done using preformulation techniques.

## 4. Conclusions

Semisolid dosage forms offer practical benefits for delivering therapeutically acceptable drug therapies directed to treating various skin diseases locally in the skin. The benefits may significantly outweigh the drawbacks of nonlocalized conventional dosage forms. Niosomal drug delivery using a simple gel has been applied for topical application to prolong drug penetration and retention into human skin. The selected desoximetasone niosomal topical gel may be used as an alternative treatment method for various skin diseases. Results showed acceptable entrapment efficiency with small niosome vesicle sizes (below 500 nm). This study provides evidence for the potential of the selected desoximetasone niosomal gel for use as a controlled release system. Key benefits include a reduction of drug concentration and dosing frequency. Niosomal gels provide novel alternatives to achieve improved patient compliance when compared to conventional topical gels. Future studies to compare this vehicle’s clinical acceptance will be needed to support this as an effective alternative to currently available therapies.

## Figures and Tables

**Figure 1 ijms-22-01535-f001:**
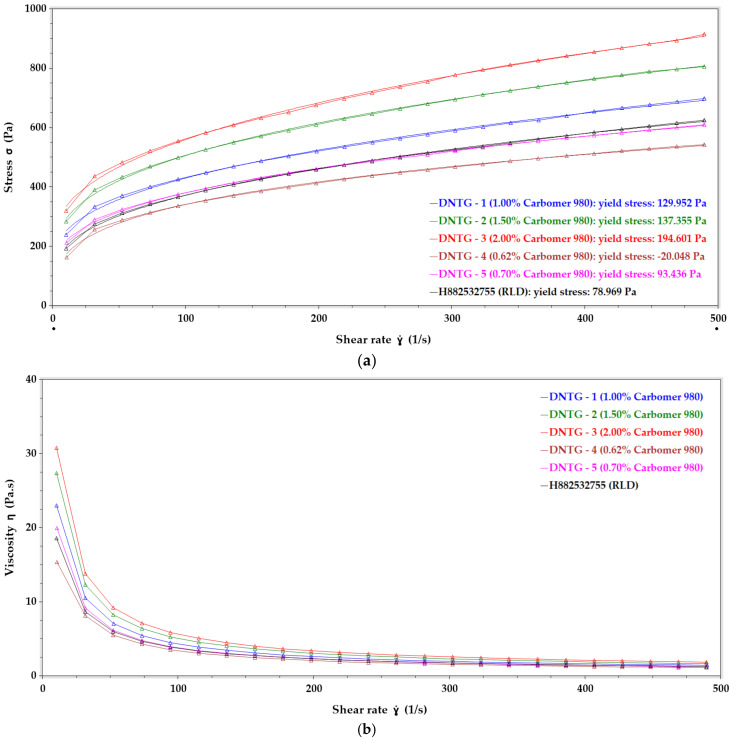
Profile overlay 1 (**a**) Yield stress and 1 (**b**) Viscosity for formulations containing Carbomer 980 concentration 0.62% (DNTG-4), 0.70% (DNTG-5), 1.00% (DNTG-1), 1.50% (DNTG-2), and 2.00% (DNTG-3), and the reference gel product (H882532755).

**Figure 2 ijms-22-01535-f002:**
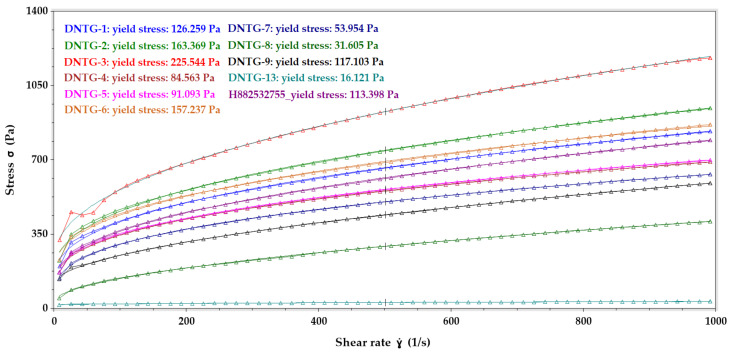
Yield stress (Herschel–Bulkley model) rheological data comparison between DNTG-1 to DNTG-9, DNTG-13 (niosomal gel), and H882532755 (reference gel).

**Figure 3 ijms-22-01535-f003:**
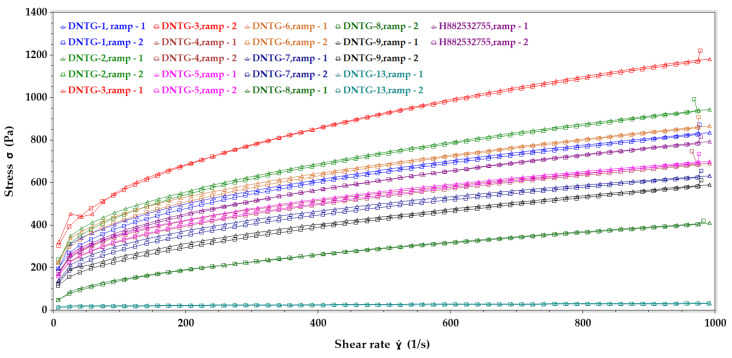
Flow curve (upward and downward curve) rheological data comparison between DNTG-1 to DNTG-9, DNTG-13 (niosomal gel), and H882532755 (reference gel).

**Figure 4 ijms-22-01535-f004:**
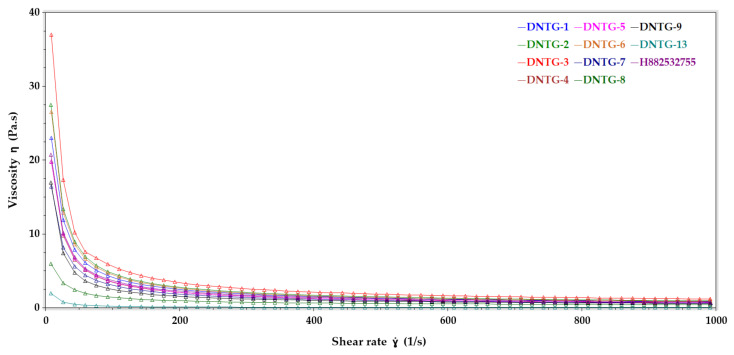
Viscosity rheological data comparison between DNTG-1 to DNTG-9, DNTG-13 (niosomal gel), and H882532755 (reference gel).

**Figure 5 ijms-22-01535-f005:**
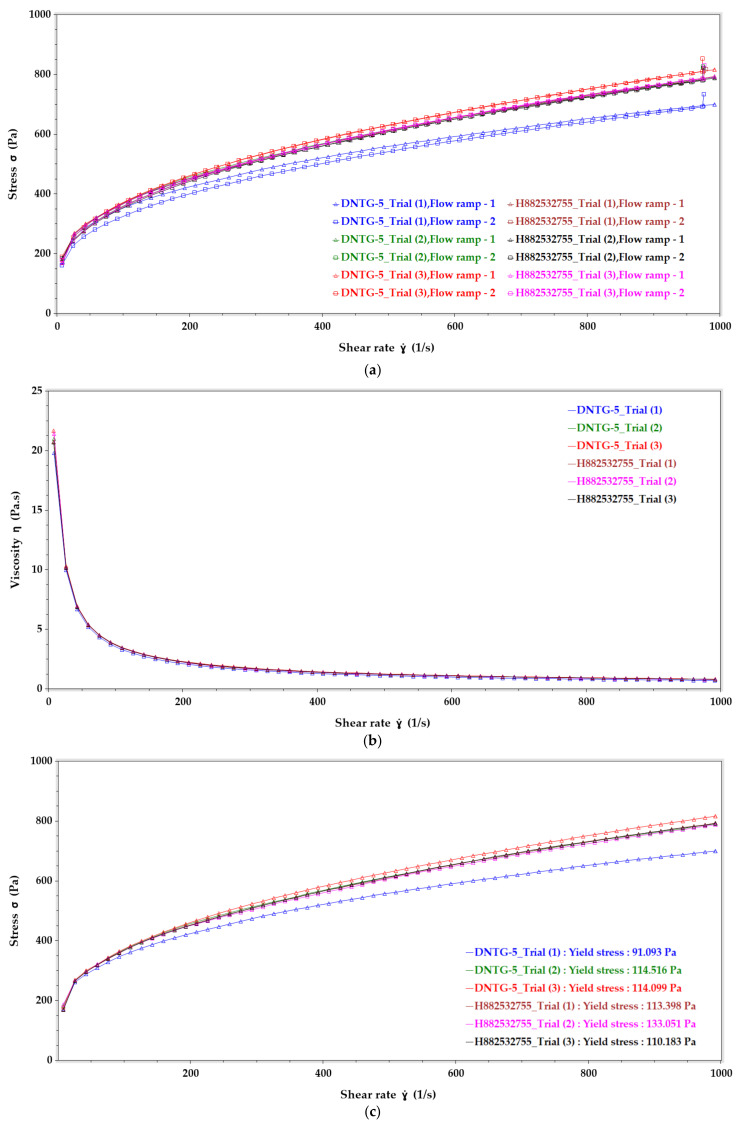
Rheological data comparison: (**a**) Flow curve (upward and downward curve), (**b**) viscosity (low to high shear rate) and (**c**) yield stress (Herschel–Bulkley model) between DNTG-5 and H882532755 (*n* = 3).

**Figure 6 ijms-22-01535-f006:**
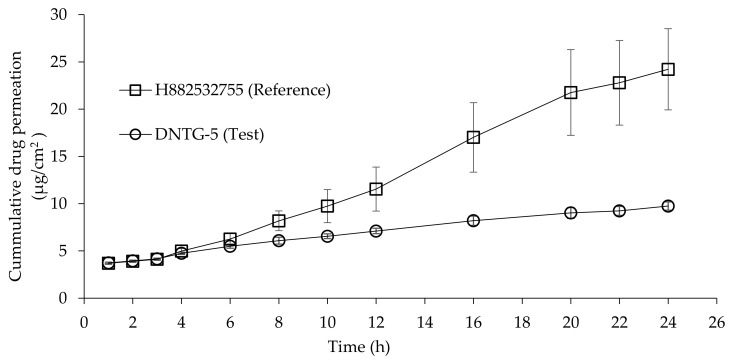
Desoximetasone permeation profile for H882532755 and DNTG-5 formulations. Time points were measured at 1, 2, 3, 4, 6, 8, 10, 12, 16, 20, 22, and 24 h. Each point represents the mean ± SD.

**Figure 7 ijms-22-01535-f007:**
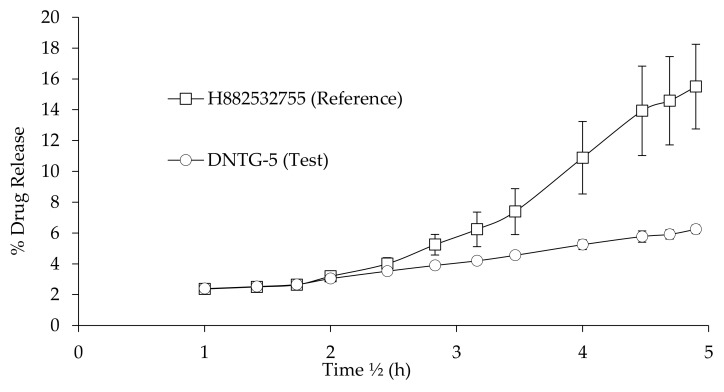
Higuchi release kinetics, comparison between reference gel and niosomal gel products (*n* = 6, mean ± SD).

**Figure 8 ijms-22-01535-f008:**
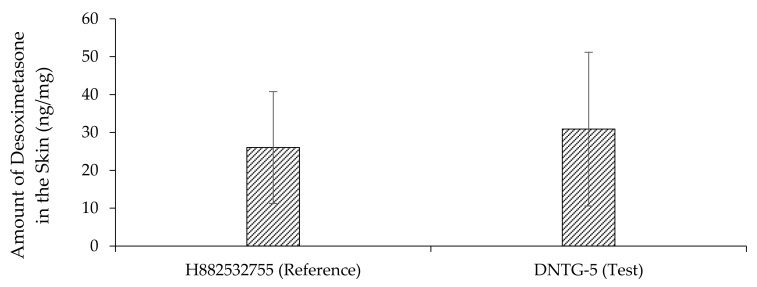
Amount of desoximetasone detected after 24 h in human cadaver skin (*n* = 6, mean ± SD) using reference gel and test niosomal gel desoximetasone formulations.

**Figure 9 ijms-22-01535-f009:**
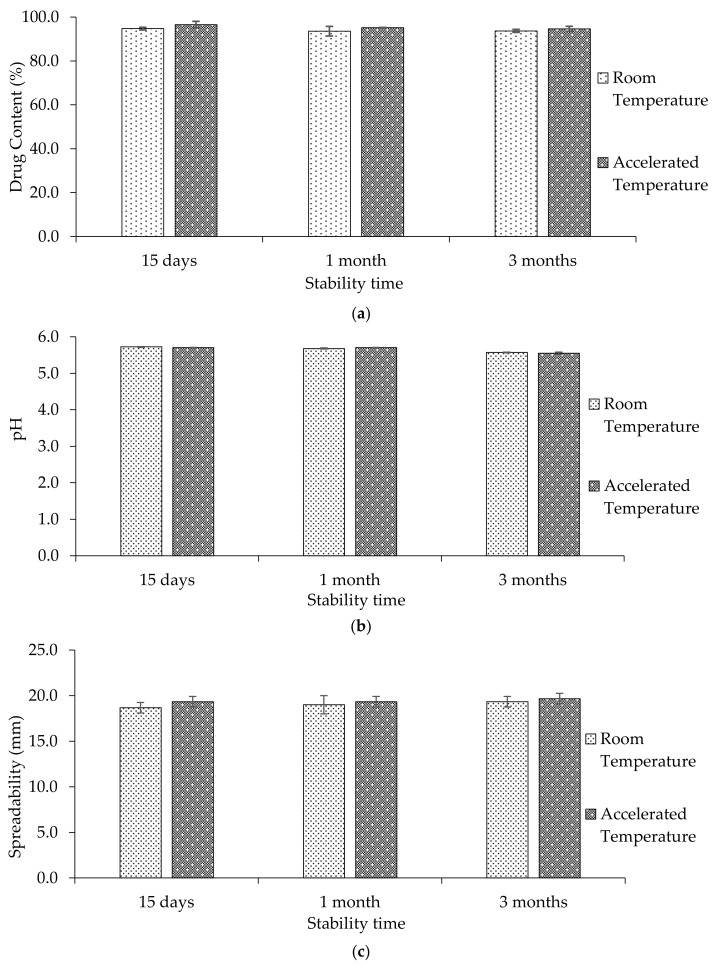
Effect of stability conditions on (**a**) drug content (%), (**b**) pH, and (**c**) spreadability (nm) of the niosomal topical gel (*n* = 3, mean ± SD).

**Table 1 ijms-22-01535-t001:** Composition of topical desoximetasone niosomal gels as DNTG (Desoximetasone Niosomal Topical Gel) (% *w*/*w*).

#	1	2	3	4	5	6	7	8	9	10	11	12	13
Materials													
Desoximetasone	0.05	0.05	0.05	0.05	0.05	0.05	0.05	0.05	0.05	0.05	0.05	0.05	0.05
Carbomer 980	1.00	1.50	2.00	0.62	0.70	-	-	-	-	-	-	-	-
Carbomer 940	-	-	-	-	-	0.70	-	-	-	-	-	-	-
Carbomer 974P	-	-	-	-	-	-	0.70	-	-	-	-	-	-
Carbomer 981	-	-	-	-	-	-	-	0.70	-	-	-	-	-
Carbomer 1342	-	-	-	-	-	-	-	-	0.70	-	-	-	-
Ethyl cellulose	-	-	-	-	-	-	-	-	-	0.70	-	-	-
Hydroxy propyl cellulose	-	-	-	-	-	-	-	-	-	-	0.70	-	-
Hydroxy propyl methyl cellulose	-	-	-	-	-	-	-	-	-	-	-	0.70	-
Xanthan gum	-	-	-	-	-	-	-	-	-	-	-	-	0.70
Edetate disodium	0.018	0.018	0.018	0.018	0.018	0.018	0.018	0.018	0.018	0.018	0.018	0.018	0.018
Docusate sodium	0.014	0.014	0.014	0.014	0.014	0.014	0.014	0.014	0.014	0.014	0.014	0.014	0.014
Transcutol	8.20	8.20	8.20	8.20	8.20	8.20	8.20	8.20	8.20	8.20	8.20	8.20	8.20
Trolamine	0.30	0.30	0.30	0.30	0.30	0.30	0.30	0.30	0.30	0.30	0.30	0.30	0.30
DI Water to QS	100.00	100.0	100.0	100.0	100.0	100.0	100.0	100.0	100.0	100.0	100.0	100.0	100.0

**Table 2 ijms-22-01535-t002:** Yield stress data for reference gel and niosomal gel formulations containing different concentrations of Carbomer 980.

Batch Detail	Yield Stress (Pa)
H882532755 (Reference gel)	78.97
DNTG-1 (1.00% Carbomer 980)	128.95
DNTG-2 (1.50% Carbomer 980)	137.36
DNTG-3 (2.00% Carbomer 980)	194.60
DNTG-4 (0.62% Carbomer 980)	−20.05
DNTG-5 (0.70% Carbomer 980)	93.44

**Table 3 ijms-22-01535-t003:** Viscosity data at low, medium, and high shear rates for reference gel and niosomal gel formulations containing different concentrations of Carbomer 980.

Batch Detail	Viscosity (Pa.s)		
	~10 Shear Rate	~240 Shear Rate	~490 Shear Rate
H882532755 (Reference gel)	18.61	2.03	1.27
DNTG-1 (1.00% Carbomer 980)	23.05	2.29	1.43
DNTG-2 (1.50% Carbomer 980)	27.28	2.69	1.66
DNTG-3 (2.00% Carbomer 980)	30.82	2.99	1.87
DNTG-4 (0.62% Carbomer 980)	15.36	1.82	1.11
DNTG-5 (0.70% Carbomer 980)	19.96	2.02	1.24

**Table 4 ijms-22-01535-t004:** Drug content, pH, spreadability, and specific gravity for desoximetasone niosomal gel and reference gel (*n* = 3, mean ± SD).

Batch Detail	Drug Content (%)	pH	Spreadability (mm)	Specific Gravity
DNTG-1	98.07 ± 0.05	5.19 ± 0.01	18.00 ± 0.00	0.957 ± 0.00
DNTG-2	95.05 ± 0.08	4.77 ± 0.00	17.33 ± 0.58	0.964 ± 0.00
DNTG-3	94.43 ± 0.95	4.39 ± 0.01	17.00 ± 0.00	0.972 ± 0.00
DNTG-4	93.03 ± 0.23	5.97 ± 0.01	19.33 ± 0.58	0.952 ± 0.00
DNTG-5	95.92 ± 0.55	5.67 ± 0.01	19.00 ± 0.00	0.954 ± 0.00
DNTG-6	98.18 ± 0.88	5.64 ± 0.01	19.00 ± 0.00	0.979 ± 0.00
DNTG-7	93.08 ± 0.96	5.80 ± 0.00	23.67 ± 2.08	0.971 ± 0.00
DNTG-8	94.69 ± 0.38	5.43 ± 0.02	26.33 ± 0.58	0.972 ± 0.00
DNTG-9	95.15 ± 0.72	5.46 ± 0.01	23.00 ± 0.00	0.972 ± 0.00
DNTG-10	94.35 ± 0.05	9.12 ± 0.01	Not available	1.007 ± 0.00
DNTG-11	96.84 ± 1.25	9.22 ± 0.00	Not available	1.009 ± 0.00
DNTG-12	101.84 ± 0.11	9.28 ± 0.02	Not available	1.008 ± 0.00
DNTG-13	97.61 ± 0.49	9.06 ± 0.01	Not available	1.006 ± 0.00
H882532755—Reference Gel	101.80 ± 0.38	5.64 ± 0.01	18.67 ± 0.58	0.947 ± 0.00

**Table 5 ijms-22-01535-t005:** Desoximetasone niosomal gel content uniformity data (*n* = 3, mean ± SD).

Batch Detail	Top Sample (%)	Middle Sample (%)	Bottom Sample (%)
DNTG-1	93.38 ± 0.05	92.60 ± 0.05	94.30 ± 0.04
DNTG-2	93.26 ± 0.18	95.31 ± 0.02	95.24 ± 0.07
DNTG-3	95.15 ± 0.86	94.66 ± 0.91	93.89 ± 0.65
DNTG-4	94.02 ± 0.47	94.33 ± 0.22	95.70 ± 0.18
DNTG-5	93.69 ± 2.01	97.06 ± 0.31	95.28 ± 7.59
DNTG-6	95.90 ± 0.43	98.13 ± 0.94	94.45 ± 0.13
DNTG-7	92.77 ± 0.62	94.41 ± 0.22	92.95 ± 0.25
DNTG-8	93.40 ± 0.97	94.67 ± 1.83	94.95 ± 0.41
DNTG-9	97.05 ± 0.82	96.61 ± 0.04	95.68 ± 1.38
DNTG-10	67.15 ± 0.90	58.21 ± 0.36	112.33 ± 0.17
DNTG-11	59.69 ± 1.32	74.77 ± 0.62	172.61 ± 0.69
DNTG-12	47.44 ± 1.53	48.46 ± 0.96	271.63 ± 0.03
DNTG-13	96.96 ± 1.81	95.50 ± 0.75	94.47 ± 1.03
H882532755—Reference Gel	101.78 ± 1.27	100.74 ± 0.34	100.88 ± 1.39

**Table 6 ijms-22-01535-t006:** Yield stress and viscosity data for niosomal gel and reference gel drug products.

Niosomal Gel and Reference Gel Rheology (Yield Stress and Viscosity) Data					
Batch Detail	Formulation Detail	Yield Stress (Pa)	Viscosity (Pa.s)		
			~8	~493	~991
DNTG-1	Carbomer 980—1.00%	126.26	23.03	1.33	0.84
DNTG-2	Carbomer 980—1.50%	163.37	27.53	1.50	0.95
DNTG-3	Carbomer 980—2.00%	225.54	37.00	1.87	1.19
DNTG-4	Carbomer 980—0.62%	84.56	19.83	1.11	0.70
DNTG-5	Carbomer 980—0.70%	91.09	19.79	1.13	0.71
DNTG-6	Carbomer 940—0.70%	157.24	26.56	1.39	0.87
DNTG-7	Carbomer 974P—0.70%	53.95	16.45	1.01	0.64
DNTG-8	Carbomer 981—0.70%	31.61	5.99	0.59	0.41
DNTG-9	Carbomer 1342—0.70%	117.00	16.98	0.89	0.59
DNTG-10 *	Ethyl cellulose—0.70%	N/Av	N/Av	N/Av	N/Av
DNTG-11 *	Hydroxy propyl cellulose—0.70%	N/Av	N/Av	N/Av	N/Av
DNTG-12 *	Hydroxy propyl methyl cellulose—0.70%	N/Av	N/Av	N/Av	N/Av
DNTG-13	Xanthan gum—0.70%	16.12	1.96	0.06	0.03
H882532755	Reference marketed gel	113.40	20.74	1.23	0.80

* Due to the intrinsic nature of the polymer, the gel was not obtained. Therefore, rheology was not performed.

**Table 7 ijms-22-01535-t007:** Physicochemical properties of desoximetasone niosomal topical gel.

Batch Detail	Color	Texture	Homogeneity	Phase Separation
DNTG-1	opaque to white	Smooth	++	No phase separation
DNTG-2	opaque to white	Smooth	++	No phase separation
DNTG-3	opaque to white	Smooth	++	No phase separation
DNTG-4	opaque to white	Smooth	++	No phase separation
DNTG-5	opaque to white	Smooth	++	No phase separation
DNTG-6	opaque to white	Smooth	++	No phase separation
DNTG-7	opaque to white	Smooth	++	No phase separation
DNTG-8	opaque to white	Smooth	++	No phase separation
DNTG-9	opaque to white	Smooth	++	No phase separation
DNTG-10	opaque to white	Liquidly	+	No phase separation
DNTG-11	opaque to white	Liquidly	+	No phase separation
DNTG-12	opaque to white	Liquidly	+	No phase separation
DNTG-13	opaque to white	Smooth	++	No phase separation
H882532755—Reference Gel	opaque to white	Smooth	++	No phase separation

+: not good; ++: good.

**Table 8 ijms-22-01535-t008:** Data comparison of desoximetasone niosomal gel and reference gel.

Test	H882532755 (Reference Gel)	DNTG-5 (Niosomal Gel)
Drug content	101.80%	95.92%
pH	5.64	5.67
Spreadability	18.67 mm	19.00 mm
Specific gravity	0.947	0.954
Content uniformity		
Top sample	101.78%	93.69 %
Middle sample	100.74%	97.06%
Bottom sample	100.88%	95.28%
**Physicochemical properties**		
Color	Opaque to white	Opaque to white
Texture	Smooth	Smooth
Homogeneity	Good	Good
Phase separation	No phase separation	No phase separation
Description	Opaque to white smooth gel	Opaque to light white smooth gel
**Rheological properties**		
Yield stress	118.88 Pa	106.57 Pa
Viscosity at low shear rate	20.95 Pa.s	20.82 Pa.s
Viscosity at medium shear rate	1.25 Pa.s	1.21 Pa.s
Viscosity at high shear rate	0.80 Pa.s	0.78 Pa.s

**Table 9 ijms-22-01535-t009:** Desoximetasone penetration through human cadaver skin up to 24 h.

Time (h)	Q up to 24 h (µg/cm^2^)	
	Lot# H882532755 (Reference Gel)	Lot# DNTG-5 (Test Gel)
1	3.70 ± 0.13	3.73 ± 0.08
2	3.91 ± 0.13	3.94 ± 0.08
3	4.12 ± 0.13	4.15 ± 0.08
4	4.98 ± 0.27	4.75 ± 0.13
6	6.25 ± 0.62	5.51 ± 0.22
8	8.19 ± 1.05	6.08 ± 0.37
10	9.74 ± 1.75	6.56 ± 0.25
12	11.54 ± 2.33	7.11 ± 0.29
16	17.01 ± 3.68	8.19 ± 0.49
20	21.76 ± 4.54	9.02 ± 0.57
22	22.79 ± 4.48	9.24 ± 0.47
24	24.22 ± 4.29	9.75 ± 0.44

Q, the cumulative amount of desoximetasone penetrated per cm^2^ up to 24 h (*n* = 6, mean ± SD).

**Table 10 ijms-22-01535-t010:** Kinetic models of desoximetasone release from the marketed gel and niosomal gel (*n* = 6, mean ± SD).

Formulation	First Order		Higuchi Model		Korsmeyer Peppas	
	Equation	(R^2^)	Equation	(R^2^)	Equation	Exponent (*n*)
H882532755	y = 0.087 x + 1.28	0.98	y = 3.661 x − 3.65	0.93	y = 0.673 x + 0.38	0.7
DNTG-5	y = 0.041 x + 1.38	0.95	y = 1.041 x + 1.03	0.99	y = 0.332 x + 0.50	0.3

**Table 11 ijms-22-01535-t011:** Amount of desoximetasone detected in human cadaver skin (*n* = 6, mean ± SD).

Skin Deposition of Desoximetasone (ng/mg)	
H882532755	DNTG-5
26.010 ± 14.8	30.88 ± 20.3

**Table 12 ijms-22-01535-t012:** Observed results for drug content, pH, spreadability for samples stored at room temperature and 40 °C temperature (*n* = 3, mean ± SD).

Stability Time	Temperature Conditions	Drug Content (%)	pH	Spreadability (mm)
Initial	N/A	93.10 ± 0.80	5.65 ± 0.01	18.67 ± 0.50
15 days	Room temperature	94.72 ± 0.62	5.72 ± 0.01	18.67 ± 0.58
1 month	Room temperature	93.57 ± 2.19	5.68 ± 0.02	19.00 ± 1.00
3 months	Room temperature	93.70 ± 0.69	5.57 ± 0.01	19.33 ± 0.58
15 days	40 °C temperature	96.58 ± 1.48	5.70 ± 0.01	19.33 ± 0.58
1 month	40 °C temperature	95.15 ± 0.15	5.70 ± 0.01	19.33 ± 0.58
3 months	40 °C temperature	94.65 ± 1.14	5.55 ± 0.03	19.67 ± 0.58

N/A = Not applicable.

**Table 13 ijms-22-01535-t013:** Physicochemical stability properties of desoximetasone niosomal gel samples.

Stability Time	Temperature Condition	Color	Phase Separation	Texture	Homogeneity *	Description
Initial	N/A	Opaque to white	No phase separation observed visually	Smooth	++	Opaque to white smooth gel
15 days	Room temperature	Opaque to white	No phase separation observed visually	Smooth	++	Opaque to white smooth gel
1 month	Room temperature	Opaque to white	No phase separation observed visually	Smooth	++	Opaque to white smooth gel
3 months	Room temperature	Opaque to white	No phase separation observed visually	Smooth	++	Opaque to white smooth gel
15 days	40 °C	Opaque to white	No phase separation observed visually	Smooth	++	Opaque to white smooth gel
1 month	40 °C	Opaque to white	No phase separation observed visually	Smooth	++	Opaque to white smooth gel
3 months	40 °C	Opaque to white	No phase separation observed visually	Smooth	++	Opaque to white smooth gel

* + is not acceptable homogeneity; ++ is acceptable homogeneity.

## Data Availability

All data reported in the manuscript.

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
