# Peer review of "Nanostructured Non-Ionic Surfactant Carrier-Based Gel for Topical Delivery of Desoximetasone"

_ijms, 2021, doi:10.3390/ijms22041535_

Round 1
Reviewer 1 Report
The manuscript demonstrates topical delivery of desoximethasone using niosomal gel formulation. The niosomal formulation was developed to optimize topical delivery of desoximetasone, and the drug content, pH, spreadability, specific gravity, content uniformity, rheology, and physicochemical properties of this formulation were analyzed. The authors found the optimal topical formulation through many experiments and compared the drug delivery efficacy with the reference gel. However, the results (figures, tables, and equations) are not well organized, some data are missing, and the explanation of the experiment is insufficient. Thus, I am sorry, but I do not recommend this manuscript to publish in the International Journal of Molecular Sciences.
- In the experimental section: Although the niosome is a major component of drug delivery in this manuscript, a description of the niosome synthesis has been omitted. The authors mentioned that niosomal formulations have already been developed in previous papers, but brief explanation should be provided so that the reader can understand how niosomes are synthesized in which substances.
- Niosome analysis data is missing. On page 18, the author claimed that the sustained release of the niosomal formulation is due to the negative charge of the niosome. But there is no data about the niosome. Also, in the conclusion section, it is written that the size of the niosome is below 500 nm. However, there is no evidence related to this in the manuscript. Thus, zeta potential and size analysis of the niosome should be conducted and presented in the manuscript.
- In my opinion, it is better to plot Figure 6 as a line graph over time rather than a bar graph to show drug permeability.
- In section 3.10.2, Kinetics analysis results, it is necessary to explain how the results were analyzed by applying them to the Higuchi model or Kosemeyer Peppas model.
- Please add error bars, standard deviations in all figures and tables, and indicate n numbers in the data.
- Tables, figures, and equations should be organized with consistent styles. In the case of figures, each text is too small. The text size is different for each table and equation. Superscript or subscript should be used properly when indicating units. Some typos and grammatical errors have also been found, so you should read the manuscript carefully and reorganize it.
Author Response
We wish to thank the reviewers for taking the time to review our manuscript. We worked to address potential concerns, and we have included additional updates to the manuscript to complete this work satisfactorily. Please see the responses to their comments in the attached file. All the additions are highlighted in red color.

Reviewer 2 Report
- Please provide the SEM to confirm the nanostructure.
- The authors should show how to encapsulate the Desoximetasone in gel.
- Please provide the drug release curve v.s. time.
- Please explain why the permeation of DNTG-5 is much lower than H882532755.
- The authors should do animal studies to prove the gel can be used for topical delivery.
Author Response

(The authors gave the same response as above.)

Round 2
Reviewer 1 Report
The manuscript is well-written. The authors carefully reflected the feedback from the reviewers. I recommend the publication of this manuscript in International Journal of Molecular Sciences. But minor points need to be corrected.
Minor comments
On page 18; Please revise Figure 2 to Figure 5.
Author Response
We wish to thank the reviewers for taking the time to review our manuscript. We worked to address potential concerns, and we have included additional updates to the manuscript to complete this work satisfactorily. Please see the responses tothe attached file.

Reviewer 2 Report
The comments have been addressed.
Author Response
Thank you for the conformation.